

# Multi-variate factorisation of numerical simulations

Daniel J. Lunt[1], Deepak Chandan[2], Harry J. Dowsett[3], Alan M. Haywood[4], George M. Lunt[5], Jonathan C. Rougier[6], Ulrich Salzmann[7], Gavin A. Schmidt[8], and Paul J. Valdes[1]

[1]School of Geographical Sciences, University of Bristol, UK
[2]Department of Physics, University of Toronto, Canada
[3]U.S. Geological Survey, USA
[4]School of Earth and Environment, University of Leeds, UK
[5]AECOM, UK
[6]School of Mathematics, University of Bristol, UK
[7]Geography and Environmental Sciences, Northumbria University
[8]NASA Goddard Institute for Space Studies, USA

**Correspondence:** Daniel J. Lunt (d.j.lunt@bristol.ac.uk)

**Abstract.** Factorisation is widely used in the analysis of numerical simulations. It allows changes in properties of a system to be attributed to changes in multiple variables associated with that system. There are many possible factorisation methods; here we discuss three previously-proposed factorisations that have been applied in the field of climate modelling: the linear factorisation, the Stein and Alpert (1993) factorisation, and the Lunt et al. (2012) factorisation. We show that, when more
than two variables are being considered, none of these three methods possess all three properties of 'uniqueness', 'symmetry', and 'completeness'. Here, we extend each of these factorisations so that they do possess these properties for any number of variables, resulting in three factorisations – the 'linear-sum' factorisation, the 'shared-interaction' factorisation, and the 'scaled-total' factorisation. We show that the linear-sum factorisation and the shared-interaction factorisation reduce to be identical. We present the results of the factorisations in the context of studies that used the previously-proposed factorisations. This
reveals that only the linear-sum/shared-interaction factorisation possesses a fourth property – 'boundedness', and as such we recommend the use of this factorisation in applications for which these properties are desirable.

## 1 Introduction

Factorisation consists of attributing the total change of some property of a system to multiple components, each component
being associated with a change to an internal variable of the system. Factorisation experiments are used in many disciplines, with early applications being in agricultural field experiments (Fisher, 1926), and widespread application in industrial and engineering design (Box et al., 2005). In this paper, we focus on factorisation of numerical model simulations of the climate system; in this case, the factorisation typically consists of attributing a fundamental property of the climate system to multiple internal model parameters and/or boundary conditions (Stein and Alpert, 1993; Lunt et al., 2012). Factorisation has been used





extensively in the climate literature; some key examples include Claussen et al. (2001), Hogrefe et al. (2004), van den Heever et al. (2006), and Schmidt et al. (2010). The factorisation proposed by Stein and Alpert (1993) has been cited more than 250 times according to Web of Science.

## 2   Previous factorisation methods

In order to introduce and discuss previous factorisation methods, we use an example case study from the field of climate
science. We turn to the Last Glacial Maximum (LGM), 21,000 years ago, the most recent time that the Earth has experienced a large-scale ice age. The LGM was 4–6 °C colder than pre-industrial (Annan and Hargreaves, 2013; Snyder, 2016); for this example, we would like to know how much of this cooling was due to a decrease in atmospheric $CO_2$ concentration and how much due to the presence of large ice sheets. In this case we would use a climate model to carry out simulations with various combinations of high and low $CO_2$ concentrations, and with and without ice sheets. In general there are interactions between
the variables so that the contributions from them do not add linearly.

It is worth at this stage introducing some notation. Here, we restrict ourselves to the case where there are two possible values for each variable, denoted '0' and '1'; having more than two values increases the computational cost of a factorisation, and can reduce the number of factors that can be assessed in a fixed computing budget. We name the fundamental property of the climate system that we are factorising as $T$. If there are $N$ variables, then the results of all possible simulations can be
uniquely identified by $T$ followed by N subscripts of either 0 or 1, with each subscript representing the value of a variable, with the variables in some predefined order. For our LGM example with two variables ($N = 2$), we have $CO_2$ (variable 1) and ice (variable 2) contributing to a global mean temperature ($T$); in this case there are 4 possible model simulations: a control simulation with pre-industrial $CO_2$ and pre-industrial ice ($T_{00}$), a second simulation with LGM $CO_2$ and pre-industrial ice ($T_{10}$), a third simulation with pre-industrial $CO_2$ and LGM ice ($T_{01}$), and an LGM simulation with LGM $CO_2$ and LGM ice
($T_{11}$) (see Figure 1a).

### 2.1   The linear factorisation

The simplest factorisation that can be carried out is a linear one. For the LGM example with 2 factors, 3 simulations are carried out in which variables are changed consecutively; for example, $T_{00}$, $T_{10}$, and $T_{11}$. The factorisation of the total change, $\Delta T$, between contributions due to $CO_2$ ($\Delta T_1$) and ice ($\Delta T_2$) would then be:

$$
\begin{aligned}
\Delta T_1 &= T_{10} - T_{00} \\
\Delta T_2 &= T_{11} - T_{10}.
\end{aligned}
\tag{1}
$$

This factorisation is illustrated graphically in Figure 1(a). However, an equally valid linear factorisation would be

$$
\begin{aligned}
\Delta T_1 &= T_{11} - T_{01} \\
\Delta T_2 &= T_{01} - T_{00},
\end{aligned}
\tag{2}
$$





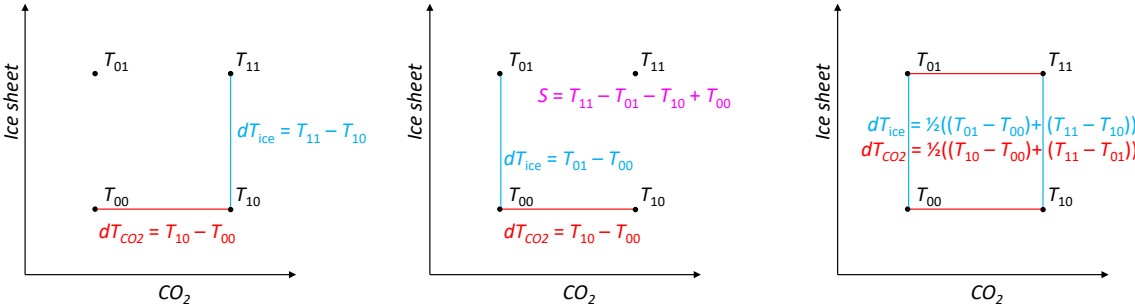

**Figure 1.** 3 different factorisation methods, for 2 variables ($CO_2$ and ice sheets). (a) Linear factorisation, (b) Stein and Alpert (1993) factorisation, (c) Lunt et al. (2012) factorisation.

and in a non-linear system this would in general give a different answer to Equation 1. In this sense, the linear factorisation method is not 'unique'. However, it is 'complete' in the sense that the individual factors sum to the total change, $\Delta T$ exactly, i.e. $\Delta T_1 + \Delta T_2 = T_{11} - T_{00}$. Considering the linear factorisation as a 'path' starting at $T_{00}$ and ending at $T_{11}$, it is also 'symmetric', in that if we instead started from $T_{11}$ we would retrieve the same numerical values for the two linear factorisations (differing just by a minus sign for the numerical value of each factor).

## 2.2 The Stein and Alpert (1993) factorisation

Stein and Alpert (1993) proposed an alternative factorisation method, illustrated in Figure 1(b). In this, for the LGM case, all four possible simulations are carried out, and the factorisation performed relative to the preindustrial case ($T_{00}$) for all variables. The non-linear terms are then all grouped together in a term which is named 'synergy', $S$:

$$\Delta T_1 = T_{10} - T_{00}$$
$$\Delta T_2 = T_{01} - T_{00}$$
$$S = T_{11} - T_{10} - T_{01} + T_{00}. \tag{3}$$

In contrast to the linear factorisation, the Stein and Alpert (1993) factorisation is unique. It is also complete because $\Delta T_1 + \Delta T_2 + S = T_{11} - T_{00}$ (in fact, $S$ is defined such that the factoristion is complete). However, it is not symmetric; if we instead performed the factorisation relative to $T_{11}$, we would in general obtain a different numerical value of the factorisation (i.e. $\Delta T_1 = T_{01} - T_{11}$).

## 2.3 The Lunt et al (2012) factorisation

Lunt et al. (2012) proposed another factorisation, in which the factorisation for a particular variable is defined as the mean of the difference between each pair of simulations that differ by just that variable. This is illustrated in Figure 1(c); the factorisation of ice is represented by the mean of the two blue lines and the factorisation of $CO_2$ is represented by the mean of the two red

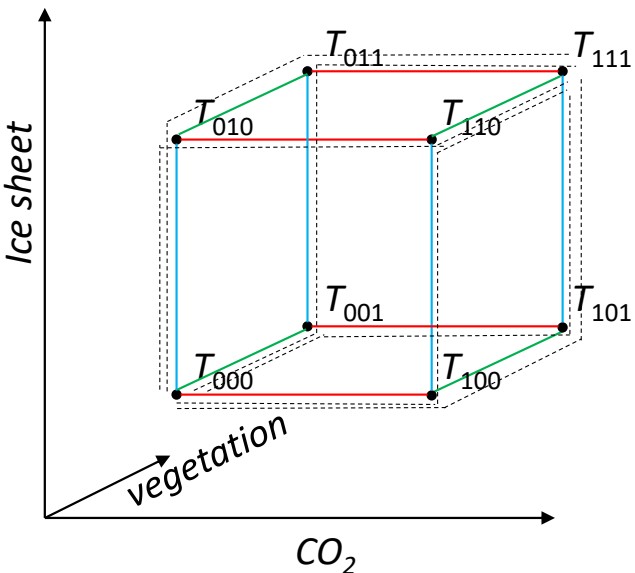

**Figure 2.** Simulations and linear factorisations in an $N = 3$ factorisation. Edges that represent changes in $CO_2$ are in red, changes in ice are in blue, and edges in vegetation are in green. The paths associated with all three possible linear factorisations are shown with dotted lines.

lines:

$$\Delta T_1 = \frac{1}{2}\{(T_{10} - T_{00}) + (T_{11} - T_{01})\}$$
$$\Delta T_2 = \frac{1}{2}\{(T_{01} - T_{00}) + (T_{11} - T_{10})\}.\tag{4}$$

The $N = 2$ factorisation in Equation 4 is unique, complete, and symmetric. It is worth noting that Equation 4 can be interpreted in multiple ways – either (i) as described above, the factorisation averages all the possible pairs of simulations that differ solely by a change in that variable, i.e. for a particular variable it is the mean of either the horizontal or vertical edges of the square in Figure 1(c); or (ii) it is the average of the two possible linear factorisations in Equations 1 and 2; or (iii) it is the average of the two possible Stein and Alpert (1993) factorisations obtained by swapping the LGM and pre-industrial (in which case the synergy terms cancel); or (iv) it is the Stein and Alpert (1993) factorisation but with the synergy term, $S$, shared equally between the two factors.

In extending to $N = 4$ variables, Lunt et al. (2012) assumed that the first of these interpretations would still hold for any number of variables. However, consider the $N = 3$ case illustrated in Figure 2, in which we have added vegetation as a third variable to contribute to LGM cooling. Averaging the edges (interpretation (i) above) would result in a factorisation:

$$\Delta T_1' = \frac{1}{4}\{(T_{100} - T_{000}) + (T_{110} - T_{010}) + (T_{101} - T_{001}) + (T_{111} - T_{011})\}$$
$$\Delta T_2' = \frac{1}{4}\{(T_{010} - T_{000}) + (T_{110} - T_{100}) + (T_{011} - T_{001}) + (T_{111} - T_{101})\}$$
$$\Delta T_3' = \frac{1}{4}\{(T_{001} - T_{000}) + (T_{101} - T_{100}) + (T_{011} - T_{010}) + (T_{111} - T_{110})\}\tag{5}$$





Although this is unique and symmetric, it is not complete, because $\Delta T'_1 + \Delta T'_2 + \Delta T'_3 \neq T_{111} - T_{000}$ (this is immediately apparent by considering the $T_{111}$ term, for which the three lines in Equation 5 sum to $\frac{3}{4}T_{111}$, whereas they are required to sum to $T_{111}$ for a complete factorisation). As such, an additional synergy term, $S$, in the sense of Equation 3, would be required for the factorisation to be complete in $N = 3$ dimensions.

### 2.4 Summary of previous factorisations

As shown above, neither the linear, or the Stein and Alpert (1993), or the Lunt et al. (2012) factorisation methods possess all three properties of uniqueness, symmetry, and completeness in $N > 2$ dimensions. These properties are often desirable in a factorisation, because any factorisation that lacks one of these properties is less easy to interpret. For example, for the LGM example above, uniqueness means that we can have a single answer to the question "why is the LGM colder than the preindustrial". Symmetry means that we obtain the same answer to the question "why is the LGM colder than the preindustrial" as to the question "why is the pre-industrial warmer than the LGM". Completeness means that we can answer the question "why is the LGM colder than the pre-industrial" by referring solely to contributions from our fundamental factors $CO_2$, ice, and vegetation, i.e. without including additional synergistic terms that are not attributed to a single factor. These synergistic terms are important and interesting, but there are cases where it can be useful or essential to only include attributable terms in the factorisation.

## 3 Extensions to the previous factorisations

Here we discuss possible extensions to the three previous factorisations discussed above, that are unique, symmetric, and complete in $N$ dimensions.

### 3.1 Extension to the linear factorisation: The linear-sum factorisation

The linear-sum factorisation arises from a generalisation to $N > 2$ dimensions of the second interpretation of Equation 4; i.e. it arises from averaging all the possible linear factorisations. This will result in a complete factorisation because each individual linear factorisation is itself complete. For three dimensions, this is illustrated by the dotted lines in Figure 2.

Each possible linear factorisation can be represented as a non-returning 'path' from the vertex $T_{000}$ to the opposite vertex $T_{111}$, traversing edges along the way (dotted lines in Figure 2). When considering the sum of all possible paths, some edges are traversed more than others. In general, those edges near the initial or final vertices are traversed more times than edges that are further away from these vertices. As such, when we average the possible linear factorisations, different edges (corresponding to different terms in the factorisation) will have different weightings. This is in contrast to Equation 5 where each term (i.e. edge of the cube) has the same weighting. For three dimensions, Figure 2 shows that the 6 edges adjacent to the initial and final





vertex are traversed twice, whereas the 6 other edges are traversed only once. Therefore, the factorisation is :

$$\Delta T_1 = \frac{1}{6} \{2(T_{100} - T_{000}) + (T_{110} - T_{010}) + (T_{101} - T_{001}) + 2(T_{111} - T_{011})\}$$

$$\Delta T_2 = \frac{1}{6} \{2(T_{010} - T_{000}) + (T_{110} - T_{100}) + (T_{011} - T_{001}) + 2(T_{111} - T_{101})\}$$

$$\Delta T_3 = \frac{1}{6} \{2(T_{001} - T_{000}) + (T_{101} - T_{100}) + (T_{011} - T_{010}) + 2(T_{111} - T_{110})\}. \tag{6}$$

This factorisation is complete ($\Delta T_1 + \Delta T_2 + \Delta T_3 = T_{111} - T_{000}$), unique, and symmetric.

To generalise to $N$ dimensions, consider an $N$-dimensional cube, which has a total of $2^N$ vertices and $N \times 2^{N-1}$ edges. There are $2^{N-1}$ edges in each dimension. There are $N!$ paths from the initial vertex of the cube to the final opposite vertex, each of which consists of a traverse along $N$ edges. Therefore, in each dimension there are a total of $N!$ edges traversed for all paths combined.

As for the 3-dimensional case above, let us label each vertex, $V$, of this $N$-dimensional cube as $V_{a_1 \cdots a_N}$, where each $a_i$ is either 0 or 1. A value $a_i = 0$ represents the first value for variable $i$, and $a_i = 1$ represents the second value for variable $i$. Each vertex is also associated with a system value, denoted $T_{a_1 \cdots a_N}$ (see Figure 2 for the case $N = 3$).

All factorisations consist of partitioning the total change, $\Delta T = T_{1 \cdots 1} - T_{0 \cdots 0}$ between N factors. Each factor is associated with a dimension, $i$, in the $N$-dimensional cube. The factorisation for dimension $i$ is $\Delta T_i$.

For the linear-sum factorisation, all paths that we consider start at the origin vertex, $0 \cdots 0$, and end at the opposite vertex $1 \cdots 1$, and are made up of a series of edges. For all edges on the $N$-dimensional cube, let us define $X$ as the set of all possible starting vertices, for a given $N$. For example, for $N = 3$, $X = \{000, 001, 010, 011, 100, 101, 110\}$. Let us define $X_i$ as the set of all possible starting vertices for an edge that is oriented in the $i$th dimension, i.e. all those vertices that have a 0 in the $i$th subscript. For example, for $N = 3$ and $i = 2$, $X_2 = \{000, 001, 100, 101\}$. Let us define $Y_i$ as the set of all possible ending vertices for an edge that is oriented in the $i$th dimension, so that $Y_i$ is related to $X_i$ by changing the $i$th subscript of each element from 0 to 1. For example, for $N = 3$ and $i = 2$, $Y_2 = \{010, 011, 110, 111\}$. Order $X_i$ and $Y_i$ so that their elements correspond. Then we write $X_i^j$ to indicate the $j$th element of $X_i$, and $Y_i^j$ as indicating the $j$th element of $Y_i$. For example, for the $X_2$ defined above, $X_2^3 = 100$.

The Lunt et al. (2012) factorisation averages along each edge oriented in dimension $i$:

$$\Delta T_i' = \frac{1}{2^{N-1}} \sum_{j=1}^{2^{N-1}} \left( T_{Y_i^j} - T_{X_i^j} \right) \tag{7}$$

For the linear-sum factorisation, we instead carry out a weighted average, with the weight for each edge in dimension $i$ given by how many times it is traversed in all $N!$ paths. Consider all the paths that traverse an edge which starts at a vertex defined by $k$ subscripts of '1' and $N - k$ subscripts of '0'. There are $k!$ possible paths to the start of this edge, and $(N - k - 1)!$ paths from the end of this edge to the final corner (defined by $N$ subscripts of '1'). Therefore, there are $k! \times (N - k - 1)!$ paths that use this edge. As such, we can write the linear-sum factorisation as:

$$\Delta T_i = \frac{1}{N!} \sum_{j=1}^{2^{N-1}} \left\{ k_i^j! \, (N - 1 - k_i^j)! \, (T_{Y_i^j} - T_{X_i^j}) \right\}, \tag{8}$$





where $k_i^j$ is the number of subscripts of '1' in $X_i^j$.

For example, for $N = 4$ and $i = 1$, we have $N! = 24$ edges traversed in this dimension, and $2^{N-1} = 8$ edges. $X_1 = \{0000, 0001, 0010, 0100, 0011, 0101, 0110, 0111\}$, and $Y_1 = \{1000, 1001, 1010, 1100, 1011, 1101, 1110, 1111\}$. For those edges with a starting subscript with $k = 0$ subscripts of '1' (i.e. 0000), the weighting $k! \, (N - 1 - k)! = 0! \, (4 - 1 - 0)! = 6$. For those edges with a starting subscript with $k = 1$ subscripts of '1' (i.e. 0001, 0010, 0100), the weighting $k! \, (N - 1 - k)! = 1! \, (4 - 1 - 1)! = 2$. For those edges with a starting subscript with $k = 2$ subscripts of '1' (i.e. 0011, 0101, 0110), the weighting $k! \, (N - 1 - k)! = 2! \, (4 - 1 - 2)! = 2$. For those edges with a starting subscript with $k = 3$ subscripts of '1' (i.e. 0111), the weighting $k! \, (N - 1 - k)! = 3! \, (4 - 1 - 3)! = 6$. Therefore, for $N = 4$ and $i = 1$, we have:

$$
\begin{aligned}
\Delta T_1 \; = \; & \frac{1}{24} \{6(T_{1000} - T_{0000}) + 2(T_{1001} - T_{0001}) + 2(T_{1010} - T_{0010}) + 2(T_{1100} - T_{0100}) + \\
& 2(T_{1011} - T_{0011}) + 2(T_{1101} - T_{0101}) + 2(T_{1110} - T_{0110}) + 6(T_{1111} - T_{0111}) \}.
\end{aligned} \tag{9}
$$

## 3.2 Extension to the Stein and Alpert (1993) factorisation: the shared-interactions factorisation

As stated in Section 2.3, the Lunt et al. (2012) factorisation for $N = 2$ can be interpreted as being identical to the Stein and Alpert (1993) factorisation but with the synergy term shared between the two factors. Here we explore what happens when this interpretation is generalised to $N > 2$ dimensions. For consistency, we use the same notation as (Stein and Alpert, 1993). In their notation, $\hat{f}_1$ represents the difference between a simulation in which only factor $i$ is modified with a simulation in which no factors are modified, and $\hat{f}_{ijk\cdots}$ represents interaction terms between the different factors. For example, for our original $N = 2$ example illustrated in Figure 1 and given in Equation 3, $\Delta T_1 \equiv \hat{f}_1$, $\Delta T_2 \equiv \hat{f}_3$, and $S \equiv \hat{f}_{12}$.

For our LGM example for $N = 3$, $\hat{f}_{12}$ is the interaction term (i.e. the synergy) between factors 1 and 2 ($CO_2$ and ice), $\hat{f}_{13}$ is the interaction term between factors 1 and 3 ($CO_2$ and vegetation), $\hat{f}_{23}$ is the interaction term between factors 2 and 3 (ice and vegetation), and $\hat{f}_{123}$ is the interaction term between all three factors. In this case, Stein and Alpert (1993) give that

$$
\begin{aligned}
\Delta T \; &= \; \hat{f}_1 + \hat{f}_2 + \hat{f}_3 + \hat{f}_{12} + \hat{f}_{13} + \hat{f}_{23} + \hat{f}_{123} \\
\hat{f}_1 \; &= \; T_{100} - T_{000} \\
\hat{f}_2 \; &= \; T_{010} - T_{000} \\
\hat{f}_3 \; &= \; T_{001} - T_{000} \\
\hat{f}_{12} \; &= \; T_{110} - (T_{100} + T_{010}) + T_{000} \\
\hat{f}_{13} \; &= \; T_{101} - (T_{100} + T_{001}) + T_{000} \\
\hat{f}_{23} \; &= \; T_{011} - (T_{010} + T_{001}) + T_{000} \\
\hat{f}_{123} \; &= \; T_{111} - (T_{110} + T_{101} + T_{011}) + (T_{100} + T_{010} + T_{001}) - T_{000}.
\end{aligned} \tag{10}
$$

As discussed in Section 2.2, this factorisation is not symmetric or unique (e.g. we could define $\hat{f}_1 = T_{011} - T_{111}$), and it is only complete if we include all the interaction terms, which are not attributed to any particular factor. By extending the interpretation of shared synergy in 2 dimensions discussed in Section 2.3, we can choose to share the interaction terms equally





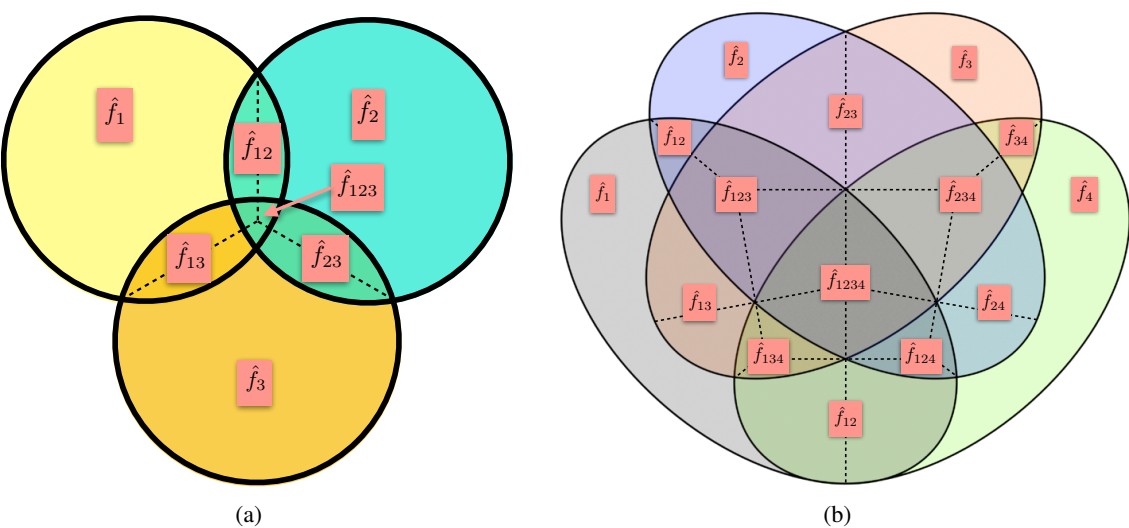

(a)            (b)

**Figure 3.** (a) Visual representation of the shared-interaction factorisation for $N = 3$, as given by Equation 10. The straight dotted lines represent the sharing of the interactions according to Equation 11. (b) Visual representation of the shared-interaction factorisation for $N = 4$. The straight dotted lines represent the sharing of the interactions according to Equation 13.

between their contributing factors, an approach applied by Schmidt et al. (2010) (although they carried out a partial factorisation in which not all combinations of all variables were included). This results in a factorisation that is complete (because we are just re-partitioning the interaction terms). It turns out that it is also symmetric. For example for $CO_2$,

$$\Delta T_1 \quad = \quad \hat{f}_1 + \frac{1}{2}\hat{f}_{12} + \frac{1}{2}\hat{f}_{13} + \frac{1}{3}\hat{f}_{123}. \tag{11}$$

This factorisation for $N = 3$ is represented visually in Figure 3(a). Equations 10 and 11 give that, for $CO_2$,

$$\Delta T_1 \quad = \quad \frac{1}{6}\left\{2(T_{100} - T_{000}) + (T_{110} - T_{010}) + (T_{101} - T_{001}) + 2(T_{111} - T_{011})\right\}. \tag{12}$$

This is identical to the equivalent term in Equation 6, indicating that the shared-interaction and linear-sum interpretations are identical for $N = 3$, and that therefore for $N = 3$ the shared-interaction factorisation is unique, symmetric, and complete.

Stein and Alpert (1993) give the generalisation of their factorisation to $N$ factors (their Equations 11-16). For $N = 4$, the

interaction terms are shared so that, for example for $CO_2$,

$$\Delta T_1 \quad = \quad \hat{f}_1 + \frac{1}{2}(\hat{f}_{12} + \hat{f}_{13} + \hat{f}_{14}) + \frac{1}{3}(\hat{f}_{123} + \hat{f}_{124} + \hat{f}_{134}) + \frac{1}{4}\hat{f}_{1234}. \tag{13}$$

This factorisation for $N = 4$ is represented visually in Figure 3(b). Again, for $N = 4$ this is the same as the linear-sum interpretation (Equation 9). We conjecture that for any $N$ these two interpretations will give identical results.





### 3.3 Extension to the Lunt et al (2012) factorisation: The scaled-total factorisation

In the scaled-total factorisation, the Lunt et al. (2012) factorisation is modified so that it is complete. This is achieved by taking the total residual term required for completeness (the 'synergy', $S$ in the sense of (Stein and Alpert, 1993)), and sharing this between the factors in proportion to the sign and magnitude of their Lunt et al. (2012) factorisation. For the $N = 3$ example of the LGM, we have that the synergy, $S$, is defined such that

$$\Delta T_1' + \Delta T_2' + \Delta T_3' + S = T_{111} - T_{000}, \tag{14}$$

where the $\Delta T_i'$ are defined in Equation 5. We then share the synergy proportionally across the three factors, such that

$$
\begin{aligned}
\Delta T_1 &= \Delta T_1' + \frac{S \Delta T_1'}{\Delta T_1' + \Delta T_2' + \Delta T_3'} \\
\Delta T_2 &= \Delta T_2' + \frac{S \Delta T_2'}{\Delta T_1' + \Delta T_2' + \Delta T_3'} \\
\Delta T_3 &= \Delta T_3' + \frac{S \Delta T_3'}{\Delta T_1' + \Delta T_2' + \Delta T_3'}
\end{aligned}
\tag{15}
$$

Equations 14 and 15 reduce to:

$$
\begin{aligned}
\Delta T_1 &= \Delta T_1' \frac{T_{111} - T_{000}}{\Delta T_1' + \Delta T_2' + \Delta T_3'} \\
\Delta T_2 &= \Delta T_2' \frac{T_{111} - T_{000}}{\Delta T_1' + \Delta T_2' + \Delta T_3'} \\
\Delta T_3 &= \Delta T_3' \frac{T_{111} - T_{000}}{\Delta T_1' + \Delta T_2' + \Delta T_3'}.
\end{aligned}
\tag{16}
$$

This shows that this factorisation can also be interpreted as simply scaling the Lunt et al. (2012) factorisation so that the sum of the factors equals $T_{111} - T_{000}$. In $N$ dimensions, this generalises to:

$$\Delta T_i = \Delta T_i' \frac{T_{1\cdots 1} - T_{0\cdots 0}}{\sum_{j=1}^{N} \Delta T_i'} \tag{17}$$

where $\Delta T_i'$ is defined in Equation 7.

For example, for $N = 4$ and $i = 1$ we have:

$$
\begin{aligned}
\Delta T_1' &= \frac{1}{8} \{ (T_{1000} - T_{0000}) + (T_{1001} - T_{0001}) + (T_{1010} - T_{0010}) + (T_{1100} - T_{0100}) + \\
&\quad (T_{1011} - T_{0011}) + (T_{1101} - T_{0101}) + (T_{1110} - T_{0110}) + (T_{1111} - T_{0111}) \} \\
\Delta T_1 &= \Delta T_1' \frac{T_{1111} - T_{0000}}{\Delta T_1' + \Delta T_2' + \Delta T_3' + \Delta T_4'};
\end{aligned}
\tag{18}
$$

and similarly for $\Delta T_2'$, $\Delta T_3'$, and $\Delta T_4'$.

## 4 Implications for previous published work

Here we discuss three examples of papers in which the Lunt et al. (2012) factorisation has been used. For each, we show how using our factorisations would affect the results in that paper.





### 4.1 Implications for Lunt et al (2012)

Lunt et al. (2012) presented a factorisation of global mean temperature change in the Pliocene (3 million years ago, the most recent time of prolonged natural global warmth relative to pre-industrial) into four variables: $CO_2$, orography, ice, and

220 vegetation. As described in Section 2.3, in extending to $N = 4$ variables, the Lunt et al. (2012) factorisation is unique and symmetric, but not complete. Using their notation, their factorisation for the $CO_2$ variable is (equivalent to Equation 9 in their paper):

$$dT'_{CO_2} = \frac{1}{8}\Big\{ (T_c - T) + (T_{oc} - T_o) + (T_{ic} - T_i) + (T_{vc} - T_v) +$$
$$(T_{ocv} - T_{ov}) + (T_{oci} - T_{oi}) + (T_{civ} - T_{iv}) + (T_{ociv} - T_{oiv}) \Big\}. \tag{19}$$

The equivalent linear-sum/shared-interaction factorisation is given by Equation 9, which in the notation of Lunt et al. (2012) is:

$$dT_{CO_2} = \frac{1}{24}\Big\{ 6(T_c - T) + 2(T_{oc} - T_o) + 2(T_{ic} - T_i) + 2(T_{vc} - T_v) +$$
$$2(T_{ocv} - T_{ov}) + 2(T_{oci} - T_{oi}) + 2(T_{civ} - T_{iv}) + 6(T_{ociv} - T_{oiv}) \Big\}, \tag{20}$$

and similarly for the other three variables.

The equivalent scaled-total factorisation is given by Equation 18, which in the notation of Lunt et al. (2012) is:

$$dT_{CO_2} = dT'_{CO_2} \frac{T_{ociv} - T}{dT'_{CO_2} + dT'_{orog} + dT'_{ice} + dT'_{veg}} \tag{21}$$

where $dT'_{CO_2}$ is given in Equation 19; and similarly for the other three variables.

In Lunt et al. (2012), although Equation 19 (Equation 9 in their paper) was presented, the four variables were actually

factorised by two $N = 2$ factorisations for all the analysis in that paper (Equation 13 in their paper). Because for $N = 2$ dimensions the Lunt et al. (2012), linear-sum/shared-interaction, and scaled-total factorisations are identical, the actual results related to Pliocene temperature change presented in Lunt et al. (2012) would not be affected by using our proposed factorisations.

### 4.2 Implications for Haywood et al (2016)

Haywood et al. (2016), in the context of the experimental design for model simulations of the Pliocene in the PlioMIP project,

presented a 3-variable factorisation of Pliocene warming into components due to $CO_2$, topography, and ice, based on the Lunt et al. (2012) factorisation (presented in their Section 3.2).

An alternative, using the linear-sum/shared-interaction factorisation that is complete, is obtained from Equation 6, which in their notation is, for $CO_2$ (and analogously for the other two components):

$$dT_{CO_2} = \frac{1}{6}\big\{ 2(E^{400} - E^{280}) + (Eo^{400} - Eo^{280}) + (Ei^{400} - Ei^{280}) + 2(Eoi^{400} - Eoi^{280}) \big\}$$

$$\tag{22}$$





Another alternative, using the scaled-total factorisation that is complete, is obtained from Equation 16, which in their notation is, for $CO_2$ (and analogously for the other two components):

$$
\begin{aligned}
dT'_{CO_2} &= \frac{1}{4}\left\{(E^{400}-E^{280})+(Eo^{400}-Eo^{280})+(Ei^{400}-Ei^{280})+(Eoi^{400}-Eoi^{280})\right\} \\
dT_{CO_2} &= dT'_{CO_2}\frac{Eoi^{400}-E^{280}}{dT'_{CO_2}+dT'_{orog}+dT'_{ice}}.
\end{aligned}
\tag{23}
$$

### 4.3 Implications for Chandan and Peltier (2018)

Chandan and Peltier (2018) applied the $N=3$ factorisation of Lunt et al. (2012) (Equation 5), as also given by Haywood et al. (2016) (first line of Equation 23), to their suite of Pliocene simulations. The factorisation was applied to each gridcell in the model, resulting in $192 \times 288 = 55,296$ factorisations over the globe. The two-dimensional mid-Pliocene minus preindustrial temperature anomaly was factorised into contributions originating from a change in $CO_2$, orography and ice sheets, and is reproduced here in Figure 4(a). Figure 4(b–d) shows the results of the original factorisation and is identical to those presented in Figure 7 of Chandan and Peltier (2018). Figure 4(f–h) shows the factorisation of the same anomaly using the linear-sum/shared-interaction method (Equations 22) while Figure 4(j–m) shows the results of employing the scaled-total method (Equations 23).

The bottom row in Figure 4 shows, for the case of each method, the residual difference between the sum of all the factors and the total change (i.e. the synergy in the sense of Stein and Alpert (1993)). The Lunt et al. (2012) method yields spatially coherent structures in the residual whose magnitude can be comparable to those of the factorized components, whereas the residuals for the other two methods are zero by definition, because they are complete (in the Figures they are very close to zero – essentially numerical noise due to round-off error). The non-linearity (indicated by the magnitude of the synergy associated with the Lunt et al. (2012) factorisation) is greatest in the North Atlantic (Figure 4d), and is likely associated with changes in the sea-ice margin that are non-linearly influenced by all three boundary conditions ($CO_2$, orography, and ice sheets).

Figure 4(j–l) reveals a problem with the scaled-total method. In these panels, the pink circles show regions where the scaled-total factorisation has very large negative or positive values for the three factors. At these locations the denominator in Eq. 16 is very small, resulting in very large magnitude positive or negative results for each factorised components, which sum to a much smaller number. This is clearly not a meaningful result (because the values in the same region in Figure 4(a) are not unusual), and although in this analysis these issues are found to occur only at isolated locations, in other cases there is potential for the problem to be more widespread. In response to this, we introduce an a fourth property of factorisations – 'boundedness'. A factorisation is bounded if the factorisation for a particular variable (e.g. $\Delta T_1$) is bounded by the minimum and maximum of all the possible single-factor factorisations for that variable. For example, for four dimensions, a factorisation is bounded if $\Delta T_1$ has a value that is not greater than the largest, or smaller than the smallest, term in Equation 18. The linear-sum/shared-interaction factorisation is by definition bounded, because it consists of a weighted average of those very terms. In contrast, the scaled-total factorisation is not bounded, and as such it should only be used with caution. It is worth noting that if absolute weightings were used in Equation 15, such that the scaled-total factorisation became (e.g. for $CO_2$):

$$
\Delta T_1 = \Delta T'_1 + \frac{S|\Delta T'_1|}{|\Delta T'_1|+|\Delta T'_2|+|\Delta T'_3|},
\tag{24}
$$



**Figure 4.** Comparison of various factorisation methods. (a) The mid-Pliocene minus preindustrial anomaly modeled by Chandan and Peltier (2017). (b–m) The top three rows present factorisations of the total anomaly into contributions arising from changes to $CO_2$ (upper, (b,f,j)), orography (middle, (c,g,k)) and ice sheets (lower, (d,h,l)), while the bottom row shows the residual ('synergy'), $T_{111} - T_{000} - (\Delta T_{CO_2} + \Delta T_{orog} + \Delta T_{ice})$, (e,i,m)). The first column (b,c,d,e) shows results using the methodology of Lunt et al. (2012) and is identical to results reported in Figure 7 of Chandan and Peltier (2018). The second column (f,g,h,i) shows results from the linear-sum/shared-interaction factorisation (Eq. 6) and the third column (j,k,l,m) shows results of the scaled-total factorisation (Eq. 16). The pink circles in the factorized results shown in the rightmost column highlight regions where the scaled-total factorisation has very large negative or positive compensating values for the three factors, due to the very small values of denominator term appearing in Eq. 16 at those locations.





then the factorisation would not result in spuriously large values (because the denominator could never approach zero). How-
ever, the factorisation would still not be bounded in our definition. For example, if $\Delta T_1'$ were negative and consisted of all
280 negative terms in the first line of Equation 5, $\Delta T_1$ could still be positive if $S$ were sufficiently large.

## 5 Conclusions

In this paper, we have reviewed three previously-proposed factorisations, and extended them to produce factorisations that are
unique, symmetric, and complete. We have presented them for 3 dimensions (i.e. 3 factors), and generalised to $N$ dimensions.
The first factorisation, 'linear-sum' (Equation 8), averages all the possible linear factorisations on the $N$-dimensional cube. The
285 second factorisation, 'shared-interaction', shares the interaction terms between each corresponding factor equally. The linear-
sum and shared-interaction factorisations are shown to reduce to be identical. The third factorisation, 'scaled-total' (Equation
17), averages all the contributions associated with the edges of the $N$-dimensional cube, and scales them by the total change
in the property being factorised. We have presented results of these extended factorisations in the context of previous work
carried out by Lunt et al. (2012), Haywood et al. (2016), and Chandan and Peltier (2018) in the context of Pliocene climate
change. This reveals that the scaled-total factorisation is not bounded, and therefore can lead to anomalous results that are hard
to interpret. Therefore we recommend the use of the linear-sum/shared-interaction factorisation for cases where the properties
of uniqueness, symmetry, and completeness, and boundedness are desirable. The properties of all the factorisations discussed
in this paper are shown in Table 1 for 2,3, and $N$ dimensions. The methods that we present here will be of particular use in the
analysis of systems with multiple variables, and have application beyond solely climate science.

*Code and data availability.* The model fields underlying Figure 4 are available from the University of Toronto Dataverse in netcdf for-
mat: https://doi.org/10.5683/SP2/QGK5B0 . The python code used to calculate the factorisations illustrated in Figure 4 is available in the
Supplement.

*Author contributions.* DJL wrote the paper. GAS made Figure 3 and DC made Figure 4. DJL devised the linear-sum factorisation, GAS
devised the shared-interaction factorisation, and GML devised the scaled-total factorisation. JCR provided the derivation of Equation 8.

*Competing interests.* The authors declare no competing interests.

*Acknowledgements.* DJL acknowledges support from NE/P01903X/1. HJD states that "Any use of trade, firm, or product names is for
descriptive purposes only and does not imply endorsement by the U.S. Government.".





**Table 1.** Properties of the factorisations discussed in this paper.

| Factorisation | Dimension | Complete | Unique | Symmetric | Bounded |
|---|---|---|---|---|---|
| Linear | 2 | ✓ | | ✓ | ✓ |
| | 3 | ✓ | | ✓ | ✓ |
| | 4 | ✓ | | ✓ | ✓ |
| Stein and Alpert (1993) | 2 | ✓ [a] | ✓ | | ✓ |
| | 3 | ✓ [a] | ✓ | | ✓ |
| | 4 | ✓ [a] | ✓ | | ✓ |
| Lunt et al. (2012) | 2 | ✓ | ✓ | ✓ | ✓ |
| | 3 | | ✓ | ✓ | ✓ |
| | N | | ✓ | ✓ | ✓ |
| scaled-total | 2 | ✓ | ✓ | ✓ | ✓ |
| | 3 | ✓ | ✓ | ✓ | |
| | N | ✓ | ✓ | ✓ | |
| linear-sum/shared-interaction | 2 | ✓ | ✓ | ✓ | ✓ |
| | 3 | ✓ | ✓ | ✓ | ✓ |
| | N | ✓ | ✓ | ✓ | ✓ |

[a] If synergy term included.

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
