# Peer review of "Multi-variate factorisation of numerical simulations"

_Geoscientific Model Development, 2020_

## Referee Comment (RC1) · Chris Brierley (Referee) · 26 Jun 2020

Review of Lunt et al. (2020)

This paper presents a meaningful step forward and an important clarification to the methodology of (paleo)climate factorisations. As such I would be happy to see it published following some small revisions.

I found the introduction of the synergy term in the Lunt et al (2012) factorisation a little bewildering at first. The manuscript didn't seem to contain an explanation as to why there is no need for a synergy term in the two-dimensional case. Is it because the averaging process effectively removes one degree of freedom (dimension)?

The re-analysis of the Chandan & Peltier paper is really nice. You do a good job of

describing the impact of the different factorisations on it. However I was surprised there was no mention as to whether this reanalysis alters the conclusions of that original paper. Perhaps you would like to comment on that.

I, like the authors, and am a climate modeller and so I'm not abreast of relevant advances in other fields. I could not help wondering if this problem had been found and addressed outside of our discipline. Would you be able to comment?

This manuscript has nine authors. Yet according to the "author contributions" statement only five have contributed to the work. Can you clarify why the others deserve authorship rather than a just credit in the acknowledgements?

Line-by-line comments:

L87. It took me a while to grasp this bracketed comment, although it actually seems quite important. Perhaps you could expand on it a little.

L118. This sentence jarred with the 3/4 aside earlier. Why is it not 3/6 by the same logic?

L158. The wrong form of citation command is used here.

L160. $f_3$ should be $f_2$

L255. Please move the reference to fig 4a slightly earlier in the sentence.

Fig. 4 Can you please label the columns with the factorisation types?

———————————————

---

## Short Comment (SC1) · 27 Jun 2020

One way around this is to explain multiple phenomena with the same set of factors. A common-mode factor may emerge, thus ruling out factors that may be used in a model but are non-contributing with the entirety of the data.

---

## Referee Comment (RC2) · Anonymous Referee #2 · 4 Oct 2020

This study is well written, and very interesting. It presents a new development to Lunt et al., (2012) investigation of the factorization method used in paleoclimate modeling. The goal is to achieve completeness, uniqueness and symmetry of the factorization, and eliminate the synergy term. Mathematically, it is a fine solution. But I do have a couple questions:

1. The completeness is achieved by averaging out different additive paths of applying climate forcings (eq. 6) or sharing the synergy term proportionally with the generated warming by the individual "forcing" factors (eq. 15). I actually quite like the synergy term, because the synergy term has the physical meaning of capturing the non-linear effects of changing vegetation, ice sheet, topo/geography and $CO_2$. Beyond what is shown in Fig. 2, additional axes are needed to capture these non-linear effects. In fact,

the points seemingly overlapping at T111 could be a visual illusion, and separable with additional axes of non-linear interactions. Wouldn't it be better to leave the synergy terms alone? Additionally, Fig. 4 is a natural result of absorbing residuals into the calculation with corrections. I am worried that the physical meaning of factorization is contaminated through this absorption instead of being enhanced. Specifically, how to interpret the differences between the left, middle and right column in a physically meaningful way? (- due to the attributions of nonlinear effect to different forcings?)

2. For LGM, whether the symmetry is a feature of climate response to CO2 forcing is questioned (e.g., Zhu et al., 2020, Clim. Past. https://doi.org/10.5194/cp-2020-86). This study shows that changes in CO2 and ice sheet have different forcing efficacies under the LGM and preindustrial climate conditions. Similarly, asymmetric vegetation, ice sheet, and CO2 forcings might be prevalent for past climates. Would it be more useful to use the proposed framework to understand the asymmetry of climate forcings and responses instead of trying to force symmetry, which might not be a real feature in climate system?

3. Lastly, this framework is described in the context of LGM, showing the LGM results would be more consistent.

---

## Author Comment (AC1) · 7 Jan 2021

We thank the two reviewers for their very helpful comments on the manuscript, which have greatly improved the paper. Here we provide a point-by-point response of our proposed changes to the manuscript. The line numbers **in blue** refer to the marked-up version of the manuscript at the end of this document, which indicates how these proposed changes would integrate into the text.

**Reviewer: Chris Brierley**

This paper presents a meaningful step forward and an important clarification to the methodology of (paleo)climate factorisations. As such I would be happy to see it published following some small revisions.

I found the introduction of the synergy term in the Lunt et al (2012) factorisation a little bewildering at first. The manuscript didn't seem to contain an explanation as to why there is no need for a synergy term in the two-dimensional case. Is it because the averaging process effectively removes one degree of freedom (dimension)?

**There is not really a "reason" why the Lunt et al (2012) factorisation is complete in two dimensions (i.e. does not require a synergy term) but is not complete for three or more dimensions (i.e. does require a synergy term) – two dimensions is just a special case. Added: "*Note that the Lunt et al (2012) factorisation is complete for N=2 without such a synergy term, but this is a case specific to N=2 as a result of cancellation of terms in Equation 4.*" See lines 111-114.**

The re-analysis of the Chandan & Peltier paper is really nice. You do a good job of describing the impact of the different factorisations on it. However I was surprised there was no mention as to whether this reanalysis alters the conclusions of that original paper. Perhaps you would like to comment on that.

**We checked the global mean values from the Chandan and Peltier study for each factorisation method and they differ by less than 10%. Added: "*The first thing to note is that the three factorisations all have very similar results; visually it is difficult to tell them apart on a regional scale, and they result in global means for each factor that differ by less than 10%. This is because, in this example, the non-linearities (i.e. the interaction terms) are relatively small. As such, the main conclusions of the Chandan and Peltier (2018) study are robust to a change in factorisation methodology.*" See lines 289-293.**

I, like the authors, and am a climate modeller and so I'm not abreast of relevant advances in other fields. I could not help wondering if this problem had been found and addressed outside of our discipline. Would you be able to comment?

**We have not found any discussion of the "problem" addressed in this manuscript in any previous work. However, we have expanded the introduction somewhat to include some more background and references to previous work: "*Factorisation experiments are used in many disciplines, with early applications being in agricultural field experiments (Fisher,1926), and widespread application in industrial and engineering design (Box et al., 2005) and other fields such as medicine (e.g. Smucker et al., 2019). The experiments that underpin such analysis are called 'factorial experiments'. In some cases, in particular when there are a large number of variables, not all combinations of all variables are tested (usually due to practical or computational limitations), and some previous work has focused on optimising the experimental design of such 'fractional factorial' experiments (e.g. Domagni et al., 2021). Furthermore, each test often has an associated error or uncertainty, and may be carried out multiple times. Analysis of such experimental designs is typically carried out using analysis of variance (ANOVA), in which the total change is represented as a model consisting of a series of 'main effects', one for each factor, and 'interaction effects' between the factors Montgomery (2013)…. In common with previously proposed factorisation methods in this field (Stein and Alpert, 1993; Lunt et al., 2012), we limit our analysis to the case where there are two possible values for*

*each variable, and where all combinations of all variables have been simulated; such an experimental design is called a $2^k$ (or two-level) full factorial experiment (Montgomery, 2013). Also in common with these studies, we assume that there is zero (or negligible) uncertainty in each simulation, which is consistent with the deterministic nature of most climate models.*" **See lines 16-33.**

This manuscript has nine authors. Yet according to the "author contributions" statement only five have contributed to the work. Can you clarify why the others deserve authorship rather than a just credit in the acknowledgements?

**We clarified the Author Contributions, which now read *"DJL led the study and wrote the first draft of the paper. GAS made Figure 3 and DC made Figure 4. DJL devised the linear-sum factorisation, GAS devised the shared-interaction factorisation, and GML devised the scaled-total factorisation. JCR provided the derivation of Equation 8. HJD, US, PJV, and AMH developed the boundary conditions and early Pliocene modelling that underpin the Pliocene simulations discussed. All authors contributed to writing the final version of the paper."* See lines 338-341.**

L87. It took me a while to grasp this bracketed comment, although it actually seems quite important. Perhaps you could expand on it a little.

**We expanded this comment, which now reads: *"This is apparent by considering the $T_{111}$ terms; the three lines in Equation 5 each include a term equal to ¼ $T_{111}$, which sum to ¾ $T_{111}$, whereas they are required to sum to $T_{111}$ for a complete factorisation. As such, an additional synergy term, S, in the sense of Equation 3, would be required for the factorisation to be complete in N=3 dimensions."* See lines 107-110.**

L118. This sentence jarred with the 3/4 aside earlier. Why is it not 3/6 by the same logic?

**Considering the $T_{111}$ terms in Equation 6, there are three lines, each of which includes a term equal to 2/6 $T_{111}$, which sum to $T_{111}$.**

L158. The wrong form of citation command is used here.
**Corrected.**

L160. f3 should be f2
**Corrected.**

L255. Please move the reference to fig 4a slightly earlier in the sentence.
**Done.**

Fig. 4 Can you please label the columns with the factorisation types?
**We will do this.**

**Interactive Comment: Paul Pukite:**

One way around this is to explain multiple phenomena with the same set of factors. A common-mode factor may emerge, thus ruling out factors that may be used in a model but are non-contributing with the entirety of the data.

**Yes, this is correct, it is possible that a factorisation may reveal that one of the factors is larger than the others, and/or that some are relatively small. In our Pliocene example from the Chandan and Peltier (2018) paper this is not the case, as ice, $CO_2$ and orography are all the same order of magnitude in the global mean. There may be some confusion here between "Factor analysis" (https://en.wikipedia.org/wiki/Factor_analysis) and "Factorisation".**

**Reviewer: Anonymous Reviewer 2:**

This study is well written, and very interesting. It presents a new development to Lunt et al., (2012) investigation of the factorization method used in paleoclimate modeling. The goal is to achieve completeness, uniqueness and symmetry of the factorization, and eliminate the synergy term. Mathematically, it is a fine solution. But I do have a couple questions:

The completeness is achieved by averaging out different additive paths of applying climate forcings (eq. 6) or sharing the synergy term proportionally with the generated warming by the individual "forcing" factors (eq. 15). I actually quite like the synergy term, because the synergy term has the physical meaning of capturing the non-linear effects of changing vegetation, ice sheet, topo/geography and CO2.

**We agree that the synergy/interaction terms can be interesting and useful in some cases. We state this in the manuscript: "*These interaction terms are important and interesting, but there are cases where it can be useful or essential to only include attributable terms in the factorisation.*". See lines 125-126.**

Beyond what is shown in Fig. 2, additional axes are needed to capture these non-linear effects. In fact, the points seemingly overlapping at T111 could be a visual illusion, and separable with additional axes of non-linear interactions. Wouldn't it be better to leave the synergy terms alone?

**In Figure 2, the lengths of the various lines do not represent the magnitude of the individual temperature changes. It would require a $4^{th}$ dimension show the temperature variable (challenging on a 2-dimensional pdf!). Similarly, in Figure 1 the temperature itself can be considered as a surface in a third dimension sitting above the 2-dimensional plane of $CO_2$ and ice sheet. We clarified this in the caption of Figure 1. However, even if we could represent this fourth dimension, then it would still be the case that the three lines in the top-right corner of Figure 2 would converge on a single point, $T_{111}$, because $T_{111}$ is single-valued. Taking two example "routes" from the bottom-left ($T_{000}$) to the top-right ($T_{111}$), the overall change is, for one route: $(T_{100}-T_{000})+(T_{101}-T_{100})+(T_{111}-T_{101})$. For another route, it is $(T_{010}-T_{000})+(T_{011}-T_{010})+(T_{111}-T_{011})$. For both of these, terms cancel and they reduce to $T_{111}-T_{000}$. This is a mathematical property, not one associated with the climate system or the presence or absence of physical nonlinearities.**

Additionally, Fig. 4 is a natural result of absorbing residuals into the calculation with corrections. I am worried that the physical meaning of factorization is contaminated through this absorption instead of being enhanced.

**If researchers are interested in the synergy terms, then they can use the Stein and Alpert (1993) factorisation, or they can use our factorisations and still calculate the synergy terms from the Stein and Alpert (1993) factorisation. Added*: "In some cases, the interaction terms may, of course, be of great interest, and in such cases a non-pure factorisation can be very informative.".* See lines 330-332.**

Specifically, how to interpret the differences between the left, middle and right column in a physically meaningful way? (- due to the attributions of nonlinear effect to different forcings?)

**The differences between the left, middle, and right columns are related to the size of the synergy terms. With larger synergy terms (strong interactions between different factors), then the three columns would be more different to each other. The fact that the three columns look very similar is because actually, in this case, the synergy terms are relatively small. Added: "*The first thing to note is that the three factorisations all have very similar results; visually it is difficult to tell them apart on a regional scale, and they result in global means for each factor that differ by less than 10%. This is because, in this example, the non-linearities (i.e. the interaction terms) are relatively small.*" See lines 289-292.**

For LGM, whether the symmetry is a feature of climate response to CO2 forcing is questioned (e.g., Zhu et al., 2020, Clim. Past. https://doi.org/10.5194/cp-2020-86). This study shows that changes in CO2 and ice sheet have different forcing efficacies under the LGM and preindustrial climate conditions. Similarly, asymmetric vegetation, ice sheet, and CO2 forcings might be prevalent for past climates. Would it be more useful to use the proposed framework to understand the asymmetry of climate forcings and responses instead of trying to force symmetry, which might not be a real feature in climate system?

**We agree that there may be (in fact, almost certainly is!) an asymmetry in forcing and/or response of the climate system when considering the transition from preindustrial to a warm climate (e.g. the Pliocene), compared to a transition from the preindustrial to a cold climate (e.g. the LGM). This is a physical property of the climate system, related to the nonlinearity and/or state dependence of forcings and feedbacks. However, the "symmetry" that we discuss and define in this paper is not a physical property, but a mathematical one. It is the symmetry between the transition from preindustrial to LGM versus LGM to preindustrial. This is a mathematical requirement (A-B = -(B-A) !) . Here we "force" this symmetry, which we believe is the correct thing to do from a mathematical viewpoint – it does not preclude analysis of the type of physical "asymmetry" that the author refers to (B-A $\neq$ -(A-C)). In fact, the reviewer is basically saying that (in our 2-dimensional) case, $T_{10}-T_{00} \neq T_{11}-T_{01}$ ; which is certainly true in general, and fully consistent with our factorisation methodologies.**

Lastly, this framework is described in the context of LGM, showing the LGM results would be more consistent.

**We agree that it makes sense to be consistent between the initial examples and the applications to the previous work. As such, we changed the LGM example to a Pliocene example, by changing the names and by changing "warmer" to "colder" and vice versa, etc. See e.g. line 40-45.**

**Other Changes:**

**We made a number of minor typographical and grammatical corrections.**

**We added a note that the Lunt et al factorisation is identical to the 'main effects' in used in ANOVA. See line 113.**

**We added the additional concept of "purity", so that we can differentiate between completeness with and without a synergy term. See e.g. lines 72-73.**

**We removed the use of the term "synergy" (apart from in introduction), ad replaced with either "interaction term" or "residual" as appropriate, for consistency with some previous work. See e.g. line 98.**

**We added some more info to some of the captions. See e.g. Figure 1,4.**

[revised manuscript text omitted]

---

## Author Response (AR2)

Please find attached our production files.